# BMAL1 Knockdown Leans Epithelial–Mesenchymal Balance toward Epithelial Properties and Decreases the Chemoresistance of Colon Carcinoma Cells

**DOI:** 10.3390/ijms22105247

**Published:** 2021-05-16

**Authors:** Yuan Zhang, Aurore Devocelle, Christophe Desterke, Lucas Eduardo Botelho de Souza, Éva Hadadi, Hervé Acloque, Adlen Foudi, Yao Xiang, Annabelle Ballesta, Yunhua Chang, Julien Giron-Michel

**Affiliations:** 1INSERM UMR-S 935, CNRS Campus, 94801 Villejuif, France; zyljxmm88@gmail.com (Y.Z.); christophe.desterke@inserm.fr (C.D.); lucasebsouza@usp.br (L.E.B.d.S.); Eva.Hadadi@vub.be (É.H.); herve.acloque@inrae.fr (H.A.); adlen.foudi@inserm.fr (A.F.); yunhua.chang-marchand@inserm.fr (Y.C.); 2Orsay-Vallée Campus, Paris-Saclay University, 91190 Gif-sur-Yvette, France; aurore.devocelle@inserm.fr; 3Institute of Life Sciences, Chongqing Medical University, Chongqing 400016, China; 4INSERM UMR-S-MD 1197/Ministry of the Armed Forces, Biomedical Research Institute of the Armed Forces (IRBA), Paul-Brousse Hospital Villejuif and CTSA Clamart, 94807 Villejuif, France; 5INSERM UMR-S 1151, Department of Immunology, Infectiology and Hematology, Institut Necker-Enfants Malades (INEM), Paris Descartes University, CNRS UMR 8253, 75730 Paris, France; yao.xiang@inserm.fr; 6INSERM UMR-S 900, Institut Curie, MINES ParisTech CBIO, PSL Research University, 92210 Saint-Cloud, France; annabelle.ballesta@curie.fr

**Keywords:** colorectal cancer, epithelial–mesenchymal transition (EMT), circadian clock, BMAL1, metastasis, chemoresistance

## Abstract

The circadian clock coordinates biological and physiological functions to day/night cycles. The perturbation of the circadian clock increases cancer risk and affects cancer progression. Here, we studied how BMAL1 knockdown (BMAL1-KD) by shRNA affects the epithelial–mesenchymal transition (EMT), a critical early event in the invasion and metastasis of colorectal carcinoma (CRC). In corresponding to a gene set enrichment analysis, which showed a significant enrichment of EMT and invasive signatures in BMAL1_high CRC patients as compared to BMAL1_low CRC patients, our results revealed that BMAL1 is implicated in keeping the epithelial–mesenchymal equilibrium of CRC cells and influences their capacity of adhesion, migration, invasion, and chemoresistance. Firstly, BMAL1-KD increased the expression of epithelial markers (E-cadherin, CK-20, and EpCAM) but decreased the expression of Twist and mesenchymal markers (N-cadherin and vimentin) in CRC cell lines. Finally, the molecular alterations after BMAL1-KD promoted mesenchymal-to-epithelial transition-like changes mostly appeared in two primary CRC cell lines (i.e., HCT116 and SW480) compared to the metastatic cell line SW620. As a consequence, migration/invasion and drug resistance capacities decreased in HCT116 and SW480 BMAL1-KD cells. Together, BMAL1-KD alerts the delicate equilibrium between epithelial and mesenchymal properties of CRC cell lines, which revealed the crucial role of BMAL1 in EMT-related CRC metastasis and chemoresistance.

## 1. Introduction

Colorectal cancer (CRC) is one of the most common cancers and represents the fourth leading cause of cancer death worldwide. In 2019, there were 145,600 new cases and approximately 51,020 deaths in the United States [1]. Metastasis in liver and lung represents one of the main causes of CRC-related mortality [2]. An insight into the molecular events driving metastasis, particularly the enhanced invasiveness and therapeutic resistance, is crucial for developing some novel treatment regimens to combat CRC. One such critical event is the epithelial-to-mesenchymal transition (EMT), an early step of CRC metastasis, by which polarized epithelial cells lose their cell polarity and cell–cell adhesion but gain mesenchymal cell properties with enhanced migratory capacity, invasiveness, metastatic potential, and drug resistance [3]. Several deregulated signaling pathways, such as Sonic Hedgehog, transformation growth factor-β (TGF-β), and Wnt pathways, are involved in the initialization and maintenance of EMT [4]. During this process, the expression of E-cadherin, referred to as the “caretaker” of the epithelial phenotype, decreases and results in the disturbance of the E-cadherin/β-catenin complex [5], which is crucial to maintaining epithelial integrity [4]. Moreover, the disturbance of this complex, ultimately, will lead to the nuclear translocation of β-catenin and transcription of EMT-promoting genes through Wnt signaling activation.

The circadian timing system (CTS) coordinates organismal behavior, including physiology and metabolism, over 24 h environmental cycles. In mammals, the suprachiasmatic nuclei (SCN) of the hypothalamus allows the coordination of circadian rhythms via peripheral molecular clocks composed of about twenty genes located in each cell of the organism. The circadian clock genes compose a complex network of auto-regulatory transcription/translation feedback loops, such as positive (BMAL1 and CLOCK) or negative (PER and CRY) loops and also feedback loops (REV-ERB α/β and ROR α/β/γ) to regulate different biological rhythm progressions [6]. The circadian clock thus directly or indirectly regulates the expression of thousands of genes in multiple cell types. Subsequently, these rhythmic regulations control many cellular processes such as autophagy, nutrient metabolism, redox regulation, cell cycle, DNA damage repair, protein folding, and cellular secretion [6,7,8,9]. An intact circadian clock is therefore essential for proper functioning of numerous physiological and behavioral processes, and its disruption may lead to disorders including cancers [10]. Indeed, higher incidence rates of colorectal, breast, and endometrial cancers have been reported in shift workers with presumed circadian disruption [11]. In addition, circadian rhythms could significantly modify the efficacy and toxicity of more than 50 anticancer drugs. Oxaliplatin, a first-line chemotherapeutic anti-colorectal cancer drug is a good example. At first, its development was halted due to the excessive toxicities, whereas further clinical chronotherapy trials in phases I, II, and III revealed its anticancer efficacy by using the optimal drug delivery time which minimized adverse effects and maximized therapeutic efficacy [12,13].

As a main positive loop element which initiates circadian oscillation, BMAL1 is considered an essential and non-redundant component of circadian clocks. Among canonical clock genes, only BMAL1 knockout results in complete loss of circadian rhythm in both SCN and peripheral tissues [14], which certificates the key role of BMAL1 in the circadian system. Studies revealed that BMAL1 is involved in the pathogenesis of human cancers, functioning either as tumor suppressor or oncogenic factor [15,16,17,18]. Although BMAL1 exhibits a globally repressive function in many tumors, a high level of BMAL1 and CLOCK expressions are often associated with poorly differentiated or late-stage CRC as well as liver metastasis [19,20]. In fact, the role of BMAL1 in CRC development remains poorly understood, in particular its link to EMT, a key process in tumor progression. Here, we evaluated in vitro how knockdown BMAL1 (BMAL1-KD) directly affects EMT as well as migration/invasion and the drug-resistance capacities of two primary human CRC cell lines (i.e., HCT116 and SW480) and a metastatic CRC cell line (i.e., SW620). Our results revealed the importance of BMAL1 in modulating the epithelial–mesenchymal equilibrium of CRC cells.

## 2. Results

### 2.1. Transcriptome of CRC Patients Shows BMAL1 Correlation to EMT and Cancer Invasiveness

Disruption of the circadian clock plays a crucial role in carcinogenesis, in particular in human CRC [11,12,13]. Different from other circadian clock gene knockouts, BMAL1 is the only gene whose deletion alone leads to complete loss of rhythmicity [14]. To evaluate the biological functions of BMAL1 during colorectal tumorigenesis, we first investigated the transcriptome of CRC patients by RNAseq in The Cancer Genome Atlas (TCGA) consortium. Starting with Z-score matrix quantification, patients were stratified with their extreme BMAL1 quantification (Z-scores over Gaussian distribution in absolute value): six patients were identified with low expression and five patients with a high expression. BMAL1 expression was found to be significantly different between these two groups of patients (*p* = 6.04 × 10^−10^, Figure 1A).

Gene set enrichment analysis (GSEA) showed a significant enrichment of epithelial–mesenchymal transition signature in BMAL1_high CRC patients as compared to BMAL1_low CRC patients (Normalized Enrichment Score: NES = −1.99, *p* < 0.001, Figure 1B). Unsupervised principal component analysis with the main EMT-enriched genes (*FUCA1, SGCB, IL15, WIPF1, TGFB1, ECM2, ADM12, POSTN, PRRX1, VIM, BASP1, and EMP3*) allowed for the discrimination of the subgroup of BMAL1-dependent CRC patients (*p* = 3.23 × 10^−5^, Figure 1C). This stratification of patients with EMT genes was confirmed by expression heatmap with hierarchical clustering (Euclidean distances, Figure 1D). Similarly, GSEA analysis showed an association between BMAL1 expression in CRC patients with multicancer invasiveness signature (NES = −2.53, *p* < 0.001, Figure 1E). Unsupervised principal component analysis with the main cancer invasiveness-enriched genes (*ASPN, GLT8D2, OLFML2B, CRISPLD2, ADAM12, POSTN, and PRRX1*) allowed for the discrimination of subgroups of BMAL1-dependent CRC patients (p = 9.9 × 10^−4^, Figure 1F). This stratification of patients with cancer invasiveness genes was confirmed by expression heatmap classification (Euclidean distances, Figure 1G). These results confirmed that BMAL1 overexpression in tumors of CRC patients was associated to EMT and cancer invasiveness program deregulation.

### 2.2. BMAL1-KD Increases E-Cadherin Expression and E-Cadherin/β-Catenin Co-Localization at the Plasma Membrane of CRC Cells

To investigate BMAL1′s involvement in the EMT process and invasiveness during colorectal tumorigenesis, two primary CRC cell lines (i.e., HCT116 and SW480) and a metastatic cell line, SW620, were transduced with lentiviruses encoding a scrambled shRNA (shScr) or a shRNA-targeting BMAL1 (shBMAL1). SW480 and SW620 cell lines are derived from the primary tumor and a metastasis site of the same patient, respectively. Previous results from our laboratory showed that BMAL1 mRNA and protein substantially decreased in the three stable BMAL1 knockdown (BMAL1-KD) cell lines compared to their proper control, thereby modulating different cellular activities such as proliferation, apoptosis, and senescence [21].

We here investigated whether BMAL1-KD could also be involved in the EMT process of colon cancer. As a main component of the cell–cell adhesion junctions, E-cadherin plays a central role in maintaining epithelium integrity [22,23,24]. Since loss of E-cadherin is a key feature of EMT in various cancers, the E-cadherin expression patterns in the three CRC cell lines were first analyzed to assess the impact of BMAL1-KD on the EMT process. Our results showed a significant increase in E-cadherin expression at mRNA (Figure 2A, RT-PCR) and protein (Figure 2B, Western blot) levels in the three BMAL1-KD cell lines compared to their respective control, except for mRNA expression in SW620 cells. This increase was particularly obvious in the BMAL1-KD SW480 cells. In addition, flow cytometry results also revealed a higher E-cadherin cell membrane expression on the BMAL1-KD CRC lines (Figure 2C).

Recruitment of β-catenin and others Catenins to E-cadherin at the cell membrane is essential in the adhesion complexes that play a crucial role for maintaining cell–cell integrity and the epithelial architecture. Disrupting E-cadherin/β-catenin complex, which conducts to a nuclear translocation of β-catenin, affects not only the cell adhesive repertoire, but also Wnt-signaling activation [25,26,27]. Thus, immunofluorescence (IF) analysis of E-cadherin (red staining) and β-catenin (blue staining) were realized to investigate if increased E-cadherin expression induced by BMAL1-KD altered membrane/nuclear localization of β-catenin in the different CRC cell lines. Consistent with previous flow cytometry analysis, a stronger staining of plasma membrane tethered E-cadherin, particularly at the interface between adjacent cells, was observed in the three BMAL1-KD CRC cell lines compared to their proper controls (Figure 2D). Moreover, an increased co-localization of β-catenin and E-cadherin at the plasma membrane was observed in HCT116 and SW480 BMAL1-KD cells. Indeed, HCT116 BMAL1-KD cells showed a predominantly cell membrane localization of β-catenin compared to its control in which β-catenin exhibited both a cell membrane and cytoplasmic localization. Interestingly, SW480 BMAL1-KD cells showed an evident plasma membrane β-catenin and E-cadherin co-localization compared to control cells in which no cortical staining of β-catenin was observed.

Western blot on cytoplasmic and nuclear extracts of BMAL1-KD and control CRC cell lines were applied to analyze the localization of β-catenin in the different cell fractions (Figure 2E). Consistently with IF experiments, a significant increase in β-catenin expression in the cytoplasmic compartments that contain cytosolic and membranous proteins was found in HCT116 and SW480 BMAL1-KD cells (Figure 2E, upper panels). In addition, both HCT116 and SW480 BMAL1-KD cells presented a significant decrease in β-catenin nuclear expression (Figure 2E, lower panels). No evident difference of β-catenin localization was observed between SW620 BMAL1-KD and control cells, neither in IF, nor in Western blot.

### 2.3. BMAL1-KD Leans Epithelial–Mesenchymal Balance of CRC Cells toward Epithelial Properties

Downregulation of E-cadherin is considered as a major hallmark of EMT as restoring E-cadherin expression is sufficient for reversal of the transformed phenotype [3]. Increased E-cadherin expression in BMAL1-KD CRC cells prompted us to check if the expression of other epithelial and mesenchymal markers were also altered. Western blot (Figure 3A) and cell surface staining in flow cytometry (Figure 3B) results revealed higher levels of EpCAM, another key epithelial denominator, in BMAL1-KD CRC cell lines as compared to controls. Moreover, the total (Figure 3A) and intracellular (Figure 3B) expressions of the epithelial marker Cytokeratin 20 (CK-20) were enhanced while those of the mesenchymal marker vimentin were decreased in BMAL1-KD CRC cell lines.

Furthermore, quantitative RT-PCR revealed a significant decrease in the expression of the mesenchymal marker N-cadherin in SW480 and SW620 BMAL1-KD cells, this gene being expressed at very low levels in HCT116 cell lines (Figure 3C). Flow cytometry analysis confirmed a decreased surface expression of N-cadherin on the three BMAL1-KD cell lines (Appendix A). Finally, quantitative RT-PCR results showed that the expression of the EMT transcription factor (EMT-TF) Twist decreased in the three BMAL1-KD CRC cell lines compared to their respective control, whereas no significant changes in the mRNA levels of the two other key EMT-TFs, Slug and Snail, were observed (Figure 3D).

### 2.4. BMAL1-KD Induces Morphological Changes in CRC Cell Lines

Cells undergoing EMT or its reverse process MET, exhibited profound morphological changes. As BMAL1-KD leans the epithelial–mesenchymal balance of the colon carcinoma cell toward epithelial properties, we investigated the cell morphology of the three CRC cell lines by phase contrast microscopy (Figure 4A). HCT116 monolayer cell cultures showed typical colonies with an epithelial cobblestone-like phenotype, whereas SW620 cells display an ovoid morphology and form small aggregates (left panels). Interestingly, HCT116 and SW620 BMAL1-KD cells appeared to grow as more densely packed clones with significantly enhanced cell-to-cell contacts (right panels). The most prominent morphological changes were especially noted in SW480 BMAL1-KD cells. Indeed, BMAL1-KD induced an obvious phenotypic change in SW480 cells which lost their mesenchymal-like fusiform morphology, characteristic of a loosely connected mesenchymal phenotype (control cells) to a more tightly associated rounded epithelial phenotype (BMAL1-KD).

Besides modified cell–cell junctions, epithelialization is associated also with actin cytoskeletal re-organization from stress fibers toward a cortical actin ring [28,29]. We then examined whether BMAL1 could remodel the actin cytoskeleton and affect F-actin distribution in these cell models. An increased formation of cortical actin rings was observed in all BMAL1-KD CRC cell lines but mostly evident in SW480 BMAL1-KD cells. Notably, F-actin was relatively evenly distributed in stress fibers in control of SW480 cells, whereas BMAL1-KD led to a decrease in cell size with an increased cortical distribution of F-actin (Figure 4B). All these results indicated that BMAL1-KD induces loss of mesenchymal traits and reinforces the epithelial phenotype in CRC cells.

### 2.5. BMAL1-KD Reduces Cell Migration and the Invasive Potential of CRC Cell Lines

As EMT promotes invasiveness of cancer cells [30], a scratch-wound healing assay and an in vitro Matrigel invasion assay were performed to check if the leaned epithelial–mesenchymal balance induced by BMAL1-KD could affect the migration and invasion properties of CRC cells. For the scratch-wound healing assay, a wound was generated in a confluent monolayer of cells. The cell’s ability to migrate into the wound area was evaluated 24 h to 72 h after the wound by phase contrast microscopy. The HCT116 and SW480 BMAL1-KD cells displayed a significantly lower capacity of migration at 48 h and 72 h than their control cell lines (Figure 5A). On the opposite, wound healing was not significantly different between the control and BMAL1-KD SW620 cells (*p* > 0.05).

For the in vitro Matrigel invasion assay, cells were added to Matrigel-coated Boyden chambers and allowed to migrate across the matrix along a serum gradient (Figure 5B). In agreement with the wound healing assay, BMAL1-KD resulted in significantly lower levels of invasion in HCT116 and SW480 cell lines. However, the SW620 BMAL1-KD cell line showed no significant differences in its invasive properties compared to its control. Taken altogether, these results suggest that the BMAL1-KD primary CRC cell lines presented decreased migratory ability and invasiveness properties compared to the controls. Consistent with reinforced cell–cell junction integrity, these results provided more evidence that BMAL1-KD leaned epithelial–mesenchymal the balance of the primary colon carcinoma cell toward epithelial properties.

### 2.6. BMAL1-KD Increases the Expression of Cell Adhesion Molecules in CRC Cell Lines

Next, we further investigated whether BMAL1-KD could influence the expression of cell adhesion molecules, such as integrin β1 (CD29), integrin alpha5 (CD49e), and CD44, that were involved in the matrix–cell adhesion [31,32] and the apico-basolateral polarity [33] of CRC cells (Figure 6). Flow cytometry analysis showed that the downregulation of BMAL1 was associated with higher membranous levels of integrin β1, integrin alpha5, and CD44 in the two primary CRC cell lines (i.e., HCT116 and SW480). However, similar membranous levels of CD44, integrin β1, and integrin alpha5 were observed in SW620 cells before and after BMAL1-KD.

### 2.7. Response to Standard Therapy Is Increased in BMAL1-KD CRC Cell Lines

In addition to the migratory and invasive abilities, recent studies have shown that the EMT phenotype is closely related to chemoresistance of cancer cells [3,34,35,36,37]. We therefore studied whether BMAL1-KD affects the resistance of CRC cells to oxaliplatin (LOH), the third-generation platinum drug widely used as a first-line chemotherapeutic agent in patients with metastatic CRC. BMAL1-KD or the control CRC cell lines were treated with different concentrations of LOH (12.5–75 µM) for 48 h (Figure 7). The MTT assay was used to evaluate cell viability after LOH treatment. The results showed that BMAL1-KD led to a distinct decreased cell viability of HCT116 and SW480 BMAL1-KD cell lines compared to their proper control. However, no differences were noted between the control and the SW620 BMAL1-KD cells.

## 3. Discussion

Epithelial-to-mesenchymal transition (EMT) is a key process driving CRC metastasis and chemoresistance. In this report, we provided evidence, for the first time, that BMAL1-KD promotes mesenchymal-to-epithelial transition (MET)-like changes of colorectal carcinoma cell lines. Consistently, several typical mesenchymal markers, such as N-cadherin and vimentin, as well as the EMT transcription factor Twist decreased, while several classic epithelial markers, such as E-cadherin, CK-20, and EpCAM, increased. Among all the molecular markers implicated in EMT, E-cadherin is referred to as the “caretaker” of the epithelial phenotype, and its loss is considered a key feature of the EMT [22,23,24]. First, E-cadherin associated with β-catenin plays a critical role in adherent junctions’ formation and epithelial integrity maintenance. In addition, the E-cadherin/β-catenin complex also plays a crucial role in modulating Wnt signaling since, by maintaining the integrity of epithelial cell–cell contacts, it allows to keep Wnt/β-catenin signals in check. Thus, disruption of the E-cadherin/β-catenin complex leads to β-catenin release and subsequent translocation to the nucleus, where it promotes the transcription of EMT-associated genes upon activation of Wnt signaling [26,27]. The nuclear accumulation of β-catenin, occurring at the invasive front of CRC, is an important hallmark of EMT [38]. Conversely, E-cadherin overexpression in cancer cells impedes their growth and metastasis [5,22]. Our study revealed that BMAL1-KD reinforced epithelial properties of CRC cells with elevated plasma membrane co-localization of E-cadherin and β-catenin as well as a decreased β-catenin nuclear location, suggesting a decreased Wnt/β-catenin pathway activity in BMAL1-KD CRC cell lines. This modification of β-catenin location is mostly evident in SW480 BMAL1-KD cell line, which could be due to two reasons: firstly, BMAL1-KD increased E-cadherin RNA and protein levels to a greater extent in SW480 when compared to HCT116 or SW620 cells. Moreover, SW480 cells express a truncated adenomatous polyposis coli (APC) protein that is deficient in its ability to promote β-catenin degradation [39,40]. Without efficient degradation induced by APC, the cytoplasmic (including cytosolic and membranous) accumulation of β-catenin is more influenced by increased E-cadherin expression and their membrane co-localization. Collectively, these elements may explain why SW480 cells presented a more evident modification of β-catenin distribution in cytoplasm and nuclear after BMAL1-KD.

Acquisition of mesenchymal phenotypes destabilizes cell–cell adhesive junctions and affects cell migratory and invasive properties [5]. Thus, the expression of cell adhesion molecules, such as E-cadherin, EpCAM, CD44, and CD166 (ALCAM), are lost at the invasive front of CRC. Our results revealed that BMAL1-KD, in addition to E-cadherin and EpCAM, increased membranous expression of several cell surface proteins implicated in the matrix–cell adhesion [31,32] and the apico-basolateral polarity [36] of CRC cells such as integrin β1 (CD29), integrin α5 (CD49e), and CD44. Thus, the reduction in migratory and invasive capacity of BMAL1-KD CRC cells should be due not only to the higher expression of epithelial key denominators (E-cadherin and EpCAM) but also because of adhesion molecules such as CD29, CD49e, and CD44. In support of this notion, it was reported that induction of CD29/CD49e clustering reduces invasion and restores apicobasolateral polarity in invasive CRC cells [33]. Moreover, CD29 plays a crucial role in supporting tumor cell attachment to the extracellular matrix. A recent study demonstrated that miR-10a silencing suppresses migration and invasion in vitro, promoting MET-like changes in SW480 cells that lead to enhanced cell–matrix adhesion induced by CD29 [31]. Moreover, inducing MET-like changes in CRC cells, BMAL1-KD reduced both migration and invasion capacities. These results are consistent with other studies showing that knocking down CLOCK, the important partner of BMAL1, led to the impairment of CRC cell migration, while CLOCK overexpression promoted their migration [41]. In addition, our immunofluorescence results in BMAL1-KD CRC cells showed a specific honeycomb-like epithelial organization of the adhesion belts delineated by E-cadherin, β-catenin, and F-actin, a phenomenon associated with the epithelialization. It was shown that BMAL1 and CLOCK could stabilize and activate RhoA [41]. Constitutive activated RhoA could enhance cortical disassembly in addition to induced actin polymerization in the cell interior [42]. Thus, BMAL1-KD alters the dynamics of F-actin/G-actin turnover and induced an increased cortical distribution of F-actin, which finally reduced cancer cell migration, and invasion. Globally, BMAL1-KD induced fewer phenotypic changes in the metastatic cell line SW620 cells and does not affect their migration and invasion capabilities compared to HCT116 and SW480 cells. Compared to the two primary CRC cell lines, metastatic SW620 cells showed very limited MET-like changes after BMAL1-KD. A likely explanation for this muted response is that SW620 cells present severely diminished core-clock gene oscillations as well as a low baseline expression of BMAL1 [21,43,44,45]. Consistent with this, our results showed also that BMAL1-KD in metastatic SW620 cells did not alter NR1D1 and CLOCK expression in contrast to our results in both primary CRC cell lines [21]. All these accumulated findings indicate that the transcription–translation feedback loops involving different circadian genes are likely to be dysfunctional in SW620 cells. Consequently, even though they were derived from the same patient, SW480 and SW620 cell lines presented only 5.5% overlap of genes with oscillating expression profiles [45]. The loss of the intact circadian clock during tumor progression probably diminished the influence of BMAL1-KD on SW620 cells compared to SW480 cells. However, only a few studies have investigated the links between the circadian timing system and the metastatic process, and there are no direct clinic proofs to show a more obvious dysfunction of transcription–translation feedback loops among circadian proteins in metastatic CRC than in primary CRC. In addition, it has been shown in vitro that a rich variety of circadian phenotypes (strong, weak, and no-oscillation of BMAL1) exist in the different CRC cell lines [43]. Moreover, it will be important to take into account the gene mutations (*KRAS, BRAF, PIK3CA, CTNNB1, BRCA2, CDKN2A, P53, APC*) present in the different CRC cell lines. All these distinct mutations, as well as the different molecular circadian clock statuses, provide unique pathological backgrounds in CRC cell lines, which could produce various cell fates after BMAL1-KD. In our study, HCT116, SW480, and SW620 cells were all mtKRAS cell lines, suggesting that our results may be valid in a context of mutated KRAS, which accounts for 35-45% of all metastatic CRC [46]. Further investigations using metastatic CRC cell lines with other mutations are required to evaluate if a link between the circadian clock and the meta-static process, in particular EMT, may exist.

By repressing the expression of the EMT-TF Twist as well as increasing EpCAM and E-cadherin at the plasma membrane of CRC cells, BMAL1-KD promoted MET-like changes, supporting the conclusion that the circadian transcription factor BMAL1 plays a pivotal role in CRC metastasis progression, regulating the balance between EMT and MET. In a previous study, we observed that BMAL1-KD enhanced PI3K/AKT/mTOR signaling that triggered different CRC cell fates (proliferation, apoptosis) [21]. While numerous studies have presented the activation of the PI3K/AKT axis as a key signaling pathway controlling EMT induction and tumor progression [47,48], BMAL1-KD favors, in the current study, the MET process of CRC cell lines. This discrepancy may be explained by the fact that EMT can be influenced either directly or indirectly by multiple signaling pathways, which cooperate with the PI3K/AKT axis, such as TGF-β, NF-κB, Ras, and Wnt/β-catenin. Although the activation of the PI3K/AKT axis could promote EMT, inhibition of Wnt/β-catenin signaling as well as a decrease in Twist expression would, on the other hand, engage BMAL1-KD cells in MET. Further studies are required to check whether other BMAL1-dependent signaling pathways are involved in the EMT process.

In recent years, circadian biology is becoming a critical factor for improving drug efficacy and diminishing drug toxicity in cancer treatment (chronotherapy) [49]. The third-generation platinum drug oxaliplatin (LOH) is widely used, in combination with 5-fluorouracil and leucovorin, as a first-line chemotherapeutic agent in patients with metastatic CRC. Although chemotherapy using LOH improves overall survival for advanced or metastatic CRC patients, intrinsic or acquired LOH-resistance is still a major factor in the poor prognosis associated with advanced CRC [50]. However, the link between the circadian cycle and drug resistance mechanisms has been the subject of only few analyses [16,51,52,53]. Among the multiple mechanisms, EMT is closely associated with LOH-resistance in CRC [54]. Indeed, CRC cells undergoing EMT induced by TGF-β treatment [36] or EMT-TF Twist overexpression [55] become resistant to LOH chemotherapy. Acquired LOH resistance can also be induced by downregulation of the EMT-related miR-200c and miR-141 [56]. On the other side, selected LOH chemoresistant CRC cells acquire an EMT phenotype [34]. However, few reports have so far linked the circadian core genes to the EMT process. For instance, PER2 may suppress EMT and tumor malignancy in breast cancer cell lines [57] and in glioma cells [58,59]. In our work, consistent with the MET induction, BMAL1-KD increased the LOH sensitivity of CRC cell lines, demonstrating that BMAL1 is implicated in EMT-mediated chemoresistance in CRC. Curiously, it was reported that BMAL1 overexpression also sensitized CRC cell lines to LOH [53]. In fact, we suppose that overexpression or knockdown of BMAL1 could disrupt the circadian oscillation pattern of BMAL1 and both influence the expression of core circadian genes and BMAL1-dependent target genes [21,60], leading, ultimately, all to an increased sensitivity to LOH treatment. All these data suggested that an appropriate BMAL1 expression level is probably a key element in affecting CRC chemoresistance to LOH.

In summary, our study showed that the circadian transcription factor BMAL1 plays crucial functions in CRC metastasis progression, promoting EMT, which is a key step in the tumors’ invasiveness and chemoresistance. Thus, therapeutic targeting of BMAL1 could be considered as a potential strategy to enhance the efficacy of chemotherapy and improve outcomes for patients with metastatic CRC.

## 4. Materials and Methods

### 4.1. RNAseq Analysis

The TCGA consortium RNAsequencing data matrix of Z-scores from the CRC cohort [61] were downloaded on the cBioPortal website [62], with access that provided associated and clinical data for each patient of the cohort. CRC patients were stratified on Z-scores of BMAL1 found in RNAseq as extremes of Gaussian distribution with 6 patients having Z-scores less than minus 1.96 (BMAL1_low) and with 5 patients having Z-scores over plus 1.96 (BMAL1_high). Gene set enrichment analysis was performed with Gene Set Enrichment Analysis (GSEA) software version 4.0.3 [63] on MsigDB version 6.2 [64]. Bioinformatics analyses were performed in R software environment version 3.5.3: boxplot with ggplot2 graph definition [65], principal component analysis with FactoMineR R package [66] and expression heatmap with pheatmap R package. Statistical significance was attested with two-sided Student’s *t*-test between groups of patients with an alpha error below 0.05.

### 4.2. Cell Culture

Human colon cancer cell lines HCT116, SW480, and SW620 were grown in Dulbecco’s modified Eagle’s medium (DMEM) supplemented with GlutaMAX (Gibco-BRL, Life Technology, Saint Aubin, France) and 10% fetal bovine serum (FBS, Hyclone, Logan, UT, USA).

### 4.3. The shRNA Cloning in Lentiviral Vector

One short hairpin RNAs (shRNAs) against human BMAL1 and a control scrambled sequence (Scr) were cloned separately into a lentiviral vector (pLKO-shBMAL1-GFP-puro and pLKO-Scr-GFP-puro), as described previously [67].

### 4.4. Lentivirus Production and Cell Transduction

Lentivirus particles were prepared and used to transduce human colon cancer cell lines HCT116, SW480, and SW620 as previously described [21,67].

### 4.5. Cytoplasmic and Nuclear Extracts Preparation

CRC cell cultures were scrapped in RIPA buffer containing protease inhibitors. Then, cytoplasmic and nuclear fractions of different cell lines were extracted by using NE-PER™ Nuclear and Cytoplasmic Extraction Reagents (Thermo Fisher Scientific, Illkirch, France) according to the manufacturer’s manual. The different extracts were stored at −80° C until Western blot analysis.

### 4.6. Western-Blot Analysis

Western blot analysis was performed with sodium dodecyl sulfate polyacrylamide gel electrophoresis (SDS-PAGE) as previously described [21]. Antibodies were used against BMAL1 (Abcam, Cambridge, UK, ab3350), E-cadherin (Cell signaling Technology, Inc., Boston, MA, USA, 14472), vimentin (Cell signaling Technology, 73260), β-catenin (Cell signaling Technology, 8480). HSC-70 (Stressgen, CA, USA, SPA-815), β-actin (Cell signaling Technology, 47778) and Lamin β1 (Abcam, ab16048) were used as control for protein loading. Image J 1.4.3 was used to quantification the western results. The *t*-test was used for statistical analysis.

### 4.7. Quantitative Real-Time PCR

Quantitative real-time PCR was performed as previous described by using LightCycler 480 SYBR Green I master kit (Roche Applied Science, Basel, Switzerland, 04707516001) [68]. Primers are listed in Appendix A. Hybridization temperature for all primers was 60 °C. The relative quantification of target RNA by using 36B4 as a reference was computed with the Relquant software (Roche, Bâle, Switzerland) with the “delta delta Ct” (ΔΔCt) method. The *t*-test was used for statistical analysis.

### 4.8. Flow Cytometry Analysis

Cells were detached with Accutase (Sigma-Aldrich, St. Quentin Fallavier, France, A6964) and cell surface expression of molecules was analyzed by flow cytometry. Briefly, cells were permeabilized or not with BD Cytofix/Cytoperm reagent (BD Pharmingen, Le Pont De Claix, France), and 105 cells were suspended in DMEM medium supplemented with 1% FCS. Indirect stainings were performed for cell surface expression of EpCAM (Bio-Techne Ltd., Lille, France, AF640), E-cadherin (Bio-Techne Ltd., AF648), N-cadherin (Abcam, ab1221), and intracellular expression of Cytokeratin-20 (CK-20, SantaCruz, Dallas, TX, USA, sc-17113) and vimentin (SantaCruz, sc-7557), using species-specific secondary antibodies conjugated to phycoerythrin. Phycoerythrin (PE)-conjugated antibodies against CD29 (Immunotools, Friesoythe, Germany, 21270294), CD49e (Immunotools, 21336491), and CD44 (Immunotools, 21270444) were used to perform direct stainings of cell surface expression markers. After 3 washes, 10,000 cells were analyzed on a LSR FortessaTM flow cytometer (BecTon Dickinson, NJ, USA), and the data were analyzed using FlowJo software (Tree Star Inc., Ashland, OR, USA). The experiment was repeated at least three times. The *t*-test was used for statistical analysis.

### 4.9. Immunofluorescence and Confocal Microscopy

Immunofluorescence stainings were performed as previously described [21] using anti-β-catenin (Cell signaling Technology, 8480) and anti-E-cadherin (Cell signaling Technology, 14472) antibodies and Hoechst 33342 (Cell signaling Technology, 4082) to stain for nuclear. The confocal images were captured by a confocal LEICA SP5-AOBS microscope with a 63X/1.4 NA oil-immersion objective.

### 4.10. Migration and Invasion Assays

Migration was assessed using a wound healing assay (scratch assay). Briefly, CRC cells were cultured in 6-well plates for 24 h to achieve 100% confluence followed by starvation in serum-free DMEM medium containing 10 μg/mL mitomycin C (Thermo Fisher Scientific, BP25312) for 2 h to completely inhibit cell proliferation. After generating a wound in the cell monolayer with a sterile 200 μL pipette tip, detached cells and cellular debris were removed three times by washing with culture medium. The cells were fed with fresh growth medium and the width of the gap between the invasion fronts was measured using Image-J software (NIH, Bethesda, MD, USA) at time points from 0 to 72 h after scratching to calculate the rate of wound closure.

Invasion assays were performed in 24-well Transwell chambers with 8 µm pore polycarbonate filter inserts (Corning, Becton Dickinson Labware, Franklin Lakes, NJ, USA). CRC cells were serum starved and exposed to mitomycin C before seeding (5 × 104) on Matrigel-coated inserts (BD Biosciences, Bedford, MA, USA) in 100 μL of serum-free medium inserts. The lower chambers were filled with 0.5 mL of complete media with 10% FBS. After 24 h and/or 48 h, assays were terminated by scraping the top of the membrane to remove non-migratory cells. The cells on the lower surface of the insert were fixed in 4% paraformaldehyde and stained with 4’,6-Diamidino-2-Phenylindole, Dihydrochloride (DAPI, Thermo Fisher Scientific, D1306) and counted. Quantification of cells was carried out by counting 10 random fields using a 10X objective. Assays were performed in triplicates.

### 4.11. Drug Sensitivity Assay

The viability of CRC cells to Oxaloplatin (LOH) was measured using the MTT assay. Cells were seeded in quadruplicate at 3 × 103 per well in 96-well plates and incubated overnight at 37 °C in a humidified environment containing 5% CO_2_. Oxaliplatin was added to the cell culture medium at various concentrations (ranging from 12.5 to 75 μM). Untreated cells were used as control. After 48 h of treatment, 20 μL of 5 mg/mL MTT (HiMedia, Levallois-Perret, France, RM1131-1G) solution was added to each well and incubated at 37 °C for 3 h. Then, the supernatant was removed and 100 μL of 1:1 solution of DMSO and methanol were added to dissolve the formazan precipitate. Absorbance of the solution was read using a microplate reader (Bio-Rad Laboratories, Hercules, CA, USA) at 540 nm, with a background correction at 655 nm. The results were analyzed using GraphPad PRISM software (GraphPad, San Diego, CA, USA) and were plotted as the mean ± SEM of the absorbance for each dose of drug tested from three independent experiments.

### 4.12. Statistical Analysis

The data were analysis by GraphPad Prism 8.0.1. Statistically significant differences were determined by *t*-test or two-way ANOVA, and *p* < 0.05 was considered to be a statistically significant result. Error bars represent standard error of the mean (±SEM) of independent experiments.

## Figures and Tables

**Figure 1 ijms-22-05247-f001:**
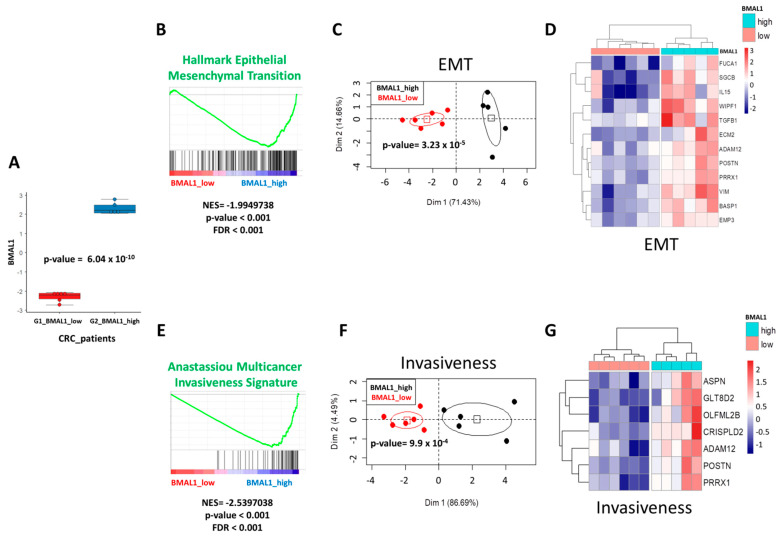
BMAL1 deregulation in colorectal carcinomas (CRC) is associated to epithelial–mesenchymal transition (EMT) and cancer invasiveness perturbations. (**A**) Boxplot of mRNA expression (Z-scores, RNAseq) in tumors of CRC with patients stratified on their BMAL1 quantification. (**B**) Enrichment for hallmark EMT gene sets in CRC tumors of patients with a high level of BMAL1 (NES: Normalized Enrichment Score). (**C**) Principal component analysis performed with EMT-enriched genes found deregulated in CRC tumor samples. (**D**) Expression heatmap of EMT genes connected to BMAL1 in CRC tumors. (**E**) Enrichment of multicancer invasiveness signature in CRC tumors of patients with a high level of BMAL1 (NES: Normalized Enrichment Score). (**F**) Principal component analysis performed with cancer invasiveness-enriched genes found deregulated in CRC tumor samples. (**G**) Expression heatmap of cancer invasiveness genes connected to BMAL1 in CRC tumors.

**Figure 2 ijms-22-05247-f002:**
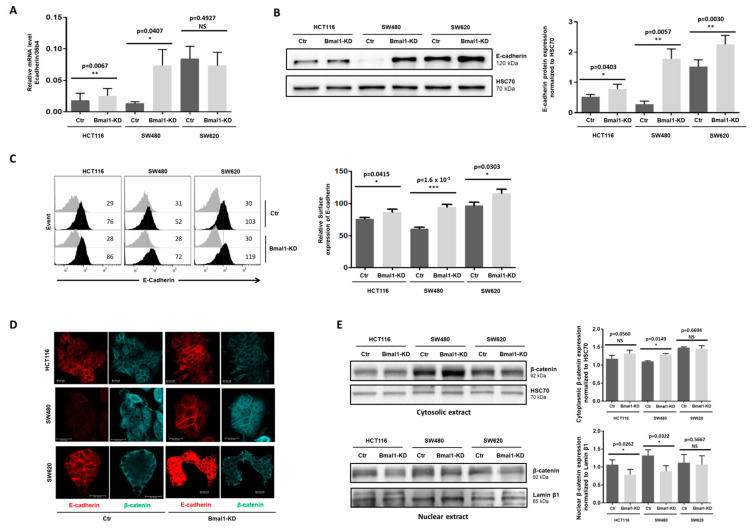
BMAL1-KD increased the plasma membrane colocation of E-cadherin and β-catenin but decreased β-catenin nuclear location. (**A**) Quantitative RT-PCR assay revealed an increased E-cadherin expression in two primary BMAL1 knockdown (BMAL1-KD) cell lines (*n* = 4, * *p* < 0.05, ** *p* < 0.01) in comparison to control (Ctr) cells. (**B**) Increased E-cadherin expression in BMAL1-KD CRC cell lines was checked by Western blot. Left, Representative Western blots of 6 independent experiments are shown. Right, Bar charts represents E-cadherin expression normalized to HSC70 (*n* = 6,* *p* < 0.05, ** *p* < 0.01). All data are shown as means ± SEM. (**C**) Enhanced E-cadherin surface expression on BMAL1-KD CRC cell lines was ascertained by flow cytometry analysis. Left, Representative staining of one experiment are shown. Cell staining with the anti-E-Cadherin antibody is represented by the dark-colored histograms and staining with the isotype-matched control antibody correspond to the gray histograms. At the right of each histogram are shown the mean fluorescence intensity values for each staining. Right, Graphs represent the relative expression of E-cadherin in three independent experiments (* *p* < 0.05, *** p < 0.001). (**D**) Confocal laser scanning microscopy showing stainings for E-cadherin (red) and β-catenin (blue). BMAL1-KD promotes elevated levels of E-cadherin and β-catenin at the plasma membrane of both HCT116 and SW480 cells, whereas BMAL1-KD only increased evidently the membrane localization of E-cadherin but not β-catenin in SW620 cells. Scale bars, 20 μm. These data are representative of three independent experiments. (**E**) Cytoplasmic (upper panels) and nuclear (lower panels) extracts of BMAL1-KD and control (Ctr) cell lines were analyzed by Western blot. Left, Representative stainings of one experiment are shown. Right, Graphs represent the quantification of Western blot signals, corresponding to β-catenin expression normalized to HSC70 (cytoplasmic extracts) or Lamin β1 (nuclear extracts). Data are represented as means ± SEM for 5 independent determinations (* *p* < 0.05). A significant increase in β-catenin cytoplasmic localization was found in SW480 BMAL1-KD cells. HCT116 BMAL1-KD cells presented an increased tendency of cytoplasmic β-catenin (*p* = 0.056). A significant decrease in β-catenin nuclear localization (*n* = 5, * *p* < 0.05) was found in the SW480 BMAL1-KD and HCT116 BMAL1-KD cell lines.

**Figure 3 ijms-22-05247-f003:**
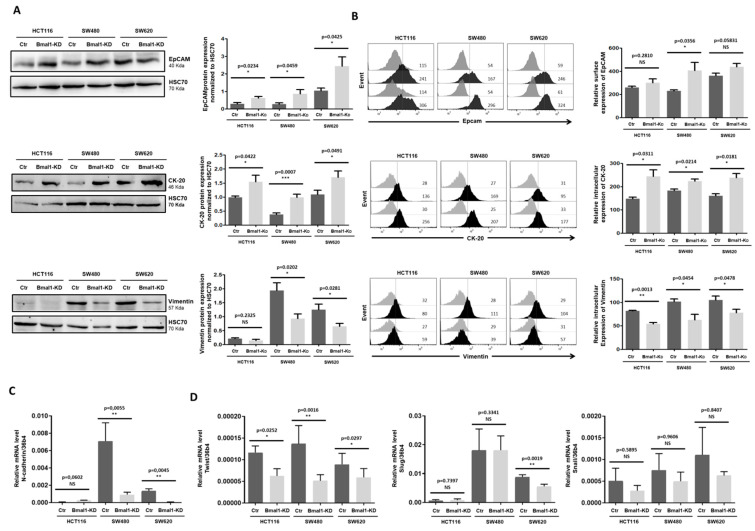
BMAL1-KD leans epithelial–mesenchymal balance of CRC cells toward epithelial properties. (**A**) Western blot revealed that BMAL1-KD increased the expression of the epithelial markers EpCAM and cytokeratin-20 (CK-20) and decreased the mesenchymal marker (vimentin) in BMAL1-KD CRC cell lines. HSC70 was used as loading controls. (**B**) EpCAM surface expression and intracellular expression of vimentin and CK-20 were analyzed by flow cytometry. Left, Representative stainings of one experiment are shown. Staining of the cells with the antigen-specific antibody is represented by the dark-colored histograms and staining with the isotype-matched control antibody correspond to the gray histograms. At the right of each histogram are shown the mean fluorescence intensity values for each staining. Right, Graphs represent the relative expression of EpCAM, CK-20, and vimentin in three independent experiments (* *p* < 0.05, ** *p* < 0.01, *** *p* < 0.001). (**C**) Quantitative RT-PCR revealed that RNA expression level of mesenchymal hallmarker N-cadherin was significantly decreased in SW480 BMAL1-KD and SW620 BMAL1-KD cell lines. The BMAL1-KD HCT116 cell line exhibited a decreased tendency of N-cadherin (*p* = 0.0762) (*n* = 5, ** p < 0.01). Data are shown as means ± SEM. (**D**) Quantitative RT-PCR analyzed the RNA expression level of three EMT-activating transcription factors Twist, Slug, and Snail in control and BMAL1-KD CRC cell lines. BMAL1-KD decreased significantly the expression of Twist but not Slug and Snail (*n* = 5, * *p* < 0.05, ** *p* < 0.01). Data are shown as means ± SEM.

**Figure 4 ijms-22-05247-f004:**
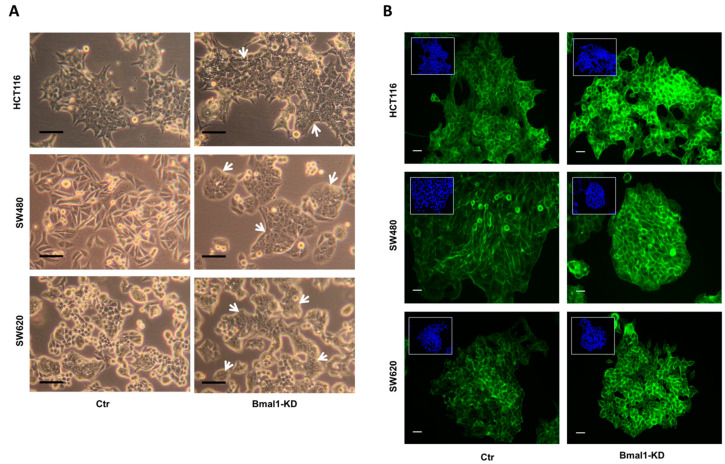
BMAL1-KD induced morphological changes on CRC cell lines. (**A**) Cells were viewed using phase contrast microscopy at the objective 20×. Compared to the control (Ctr) cells, CRC cells formed densely packed clones in the three BMAL1-KD CRC cell lines (white arrows). Morphological changes were most prominent in SW480 BMAL1-KD cells, with loss of an elongated and spindle-shaped morphology and acquisition of a typical epithelial shape with a cobblestone-like morphology. Scale bars, 50 μm. (**B**) F-actin distribution profile was analyzed by fluorescence microscopy with phalloidin (green). Nuclei staining with DAPI (blue) was inserted at the top-left of each image. Scale bars, 15 μm. The F-actin distribution profile was similar to those of E-cadherin and β-catenin in BMAL1-KD CRC cells, revealing a specific honeycomb-like epithelial organization of the adhesion belts delineated by E-cadherin, β-catenin, and F-actin.

**Figure 5 ijms-22-05247-f005:**
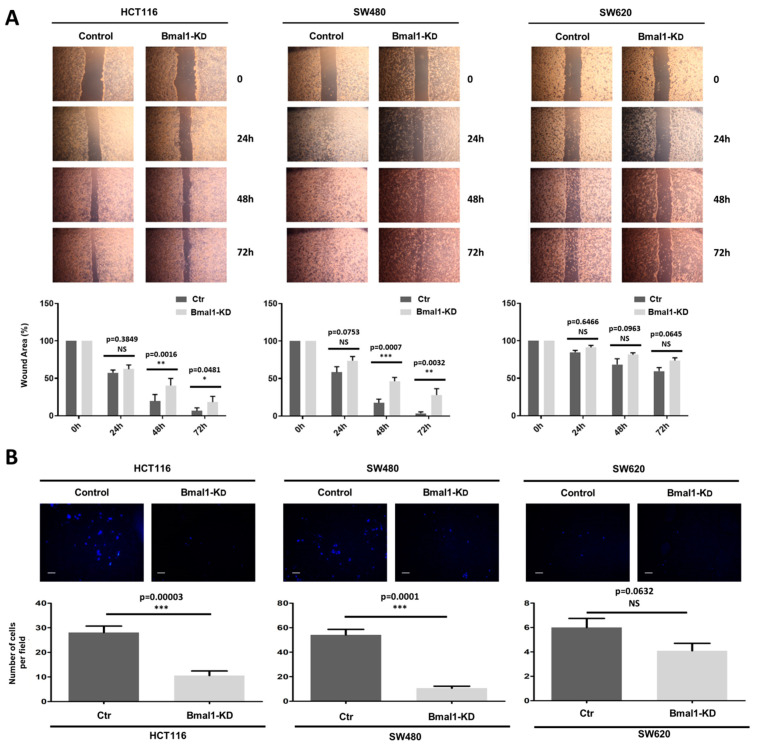
BMAL1-KD inhibits cell migration and invasion of CRC cell lines. (**A**) A scratch-wound healing assay was applied for cell migration assay. Top, Artificial wounds were made in confluent monolayers of control and BMAL1-KD CRC cell lines. Migration of CRC cells towards the wound, photographed from 24 h to 72 h. The top panel shows one of three independent experiments. Bottom, HCT116 BMAL1-KD and SW480 BMAL1-KD cells displayed lower levels of migratory activity in comparison to control cell lines (*n* = 3, * *p* < 0.05, ** *p* < 0.01, *** *p* < 0.001). No significant differences in migratory activity were observed between BMAL1-KD and control SW620 cells. Data are shown as means ± SEM. (**B**) Invasion assay. Top, The invasive potential of BMAL1-KD CRC cell lines and their control were analyzed in Matrigel Transwells after 96 h by fluorescence microscopy. DAPI was used to stain the nuclei and to determine the number of invasive cells. A representative image of random fields is shown for each cell type. Scale bar, 50 μm. Bottom, Graphs represent the mean of the number of invaded cells per field in three independent experiments. Only the primary BMAL1-KD CRC cells had a lower invasion capacity than control cells (*n* = 3, *** *p* < 0.001). Data are shown as means ± SEM.

**Figure 6 ijms-22-05247-f006:**
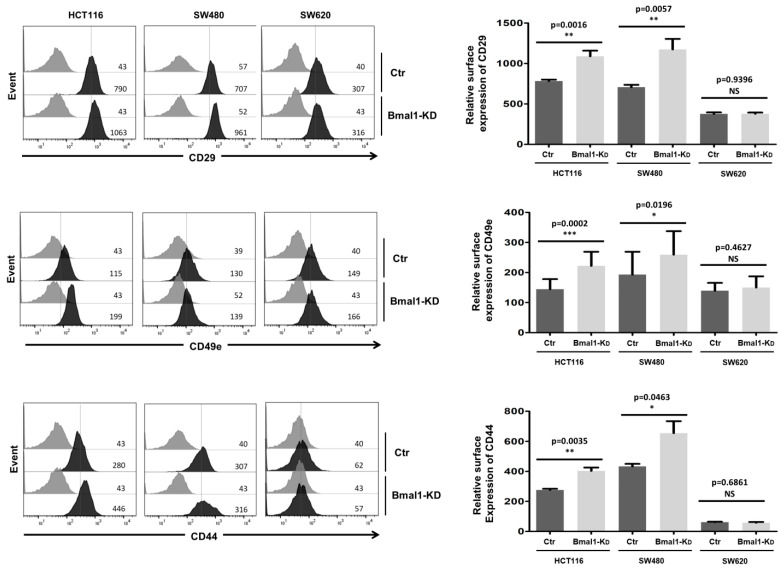
BMAL1-KD increased the expression of cell adhesion molecules on CRC cell lines. Flow cytometry analysis of surface expression of CD29 (integrin β1), CD49e (integrin α5), and CD44 adhesion molecules. Left, a representative direct staining of four independent experiments is shown. The dark-colored bars correspond to cells incubated with phycoerythrin (PE)-conjugated antibodies against CD29, CD49e, and CD44, and grey bars correspond to cells incubated with the isotype-matched control antibody. Mean fluorescence intensity values for each marker are shown at the right of each histogram. Right, BMAL1-KD led to increased expression of membranous CD29, CD49e, and CD44, mostly in primary BMAL1-KD CRC cell lines (*n* = 3, * *p* < 0.05, ** *p* < 0.01, *** *p* < 0.001). Data are shown as means ± SEM.

**Figure 7 ijms-22-05247-f007:**
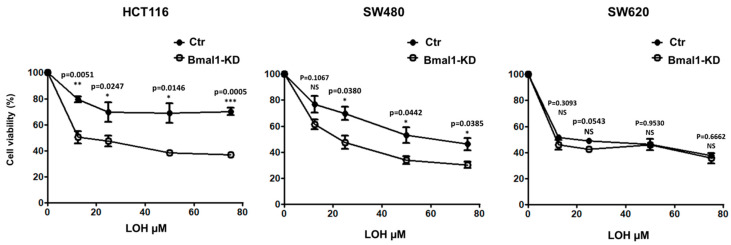
BMAL1-KD sensitizes CRC cell lines to oxaliplatin treatment. The sensitivity of CRC cells to oxaliplatin (LOH) drug was evaluated by the MTT cell viability assay. BMAL1-KD and their control cells were treated for 24 h to 48 h with increasing concentrations of LOH (from 12.5 to 75 μM). Cell viability after drug treatment is presented as a percentage relative to untreated cells. The mean values for three experiments are shown; error bars correspond to 95% confidence intervals. BMAL1-KD led to a distinct decrease in cell viability of HCT116 and SW480 cell lines under LOH treatment compared to their control cells, whereas no significant differences were observed with the metastasis SW620 cell line (*n* = 3, * *p* < 0.05, ** *p* < 0.01, *** *p* < 0.001). Data are shown as means ± SEM.

## Data Availability

The data presented in this study are available upon request from the corresponding author.

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
