# Peer review of "BMAL1 Knockdown Leans Epithelial–Mesenchymal Balance toward Epithelial Properties and Decreases the Chemoresistance of Colon Carcinoma Cells"

_ijms, 2021, doi:10.3390/ijms22105247_

Round 1

Reviewer 1 Report

The manuscript by Zhang et al reports data on how BMAL1 knockdown influences the epithelial-mesenchymal balance and what therapeutical consequences its downregulation confers in colorectal cancer. The study is valuable, the manuscript is carefully prepared and the text is easy-to-follow. One major limitation is that the authors only investigated three cell lines in total. The analysis of some patient samples could have provided added value.

Minor comments:

  • Could the authors include some more discussion about why the SW620 cell line showed different results in several tests as compared to the other two (primary) cell lines? Can we expect similar data in other metastatic cell lines? 
  • Decimals should be displayed with dots instead of commas in all figures.
  • Fig 4 legend: was the magnification or the objective x20?
  • cBioPortal instead of Cbioportal

Reviewer 2 Report

The article with the title “BMAL1 knockdown leans epithelial-mesenchymal balance to- ward epithelial properties and decreases the chemoresistance of colon carcinoma cells” is in generally well done, but I would offer these comments to the investigators: 

  • Τhere are some minor language errors.
  • There are some minor grammatical errors.
  • The authors used HCT116 a mtKRAS CRC cell line with EMT phenotype.
  • All three CRC cell lines bearing mtKRAS. I recommend to design an experiment with a CRC cell line with mtBRAF and EMT phenotype such as RKO.
  • The authors mentioned that circadian clock affect several cellular process including autophagy. Autophagy as a basic homeostatic mechanism regulates different sub-cellular aspect such as protein and endosomal trafficking. I strongly recommend to investigate the protein levels of some autophagy markers such as LC3B, p62 or Beclin-1 after BMAL1knockdown and oxaliplatin treatment.
